# Diffusion in translucent media

Zhou Shi[1,2] & Azriel Z. Genack[1]

Diffusion is the result of repeated random scattering. It governs a wide range of phenomena from Brownian motion, to heat flow through window panes, neutron flux in fuel rods, dispersion of light in human tissue, and electronic conduction. It is universally acknowledged that the diffusion approach to describing wave transport fails in translucent samples thinner than the distance between scattering events such as are encountered in meteorology, astronomy, biomedicine, and communications. Here we show in optical measurements and numerical simulations that the scaling of transmission and the intensity profiles of transmission eigenchannels have the same form in translucent as in opaque media. Paradoxically, the similarities in transport across translucent and opaque samples explain the puzzling observations of suppressed optical and ultrasonic delay times relative to predictions of diffusion theory well into the diffusive regime.

[1] Department of Physics, Queens College and Graduate Center of the City University of New York, Flushing, NY 11367, USA. [2] Chiral Photonics Inc., 26 Chapin Road, Pine Brook, NJ 07058, USA. Correspondence and requests for materials should be addressed to A.Z.G. (email: genack@qc.edu)

**E**instein showed that microscopically visible particles buffeted by stochastic molecular forces perform a random walk that can be described by the diffusion equation once the initial motion of particles is randomized[1]. The diffusion approach also describes the transport of classical and quantum waves in multiply scattering media[2–20]. Waves entering a static disordered sample interfere to produce a wavelength-scale speckled pattern of energy or particle density that is a unique fingerprint of the wave interaction with the disordered sample. When such patterns are averaged over a large ensemble of statistically equivalent samples, a smoothed profile of energy density results that is a solution of the diffusion equation[6]. The diffusion approach is assumed to fail, on time scales shorter than the scattering time[9] and on length scales smaller than the transport mean free path, $\ell$[1], in which the particle direction is randomized. On these scales, it is assumed that transport can only be described by a detailed accounting of radiative transfer within the sample[2, 20].

The transmission of waves through a disordered material is fully characterized by the transmission matrix, $t$, whose elements $t_{ba}$ are the field transmission coefficients between complete sets of $N$ orthogonal propagating channels on each side of the sample[21–33]. For an incident field in channel $a$, $E_a$, the transmitted field in channel $b$, $E_b$, can be expressed as the sum of the coherent field, with the same intensity pattern as the incident field, and a random field, which is uncorrelated with $E_a$, $E_b = E_{\text{coherent}} + E_{\text{random}} = \langle t_{ba}\rangle E_a\delta_{ab} + \delta E_b$. Here $\langle\cdots\rangle$ represents the average over random sample configurations and $\delta_{ab} = 1$ for $a = b$, and 0 otherwise.

A widely held view is that transport in the translucent and diffusive sample regimes regimes is totally dissimilar. True, diffusion is built from a series of random ballistic steps. However, the wave retains a degree of spatially coherence during each step, whereas multiply scattered waves are randomized with vanishing correlation across the sample. As a result, many characteristics of transport are totally different in these two regimes, as is illustrated in the next section, and propagation is described using different formalisms.

In this article, we explore the relationship between wave propagation in translucent and diffusive samples. Here we show that, notwithstanding the stark differences between transport in translucent and opaque samples, the underlying structure of transport is strikingly similar. The scaling of transmission and the energy density inside a random medium illuminated by random waveforms have identical forms. The energy density inside the sample falls linearly and extrapolates to zero at the same distance beyond the sample in both regimes. At the same time, the average energy density profiles in the interior of specific transmission eigenchannels have nearly identical forms. We show that the source of these similarities is the correlation within the transmission matrix, which leads to characteristic repulsion between transmission eigenvalues on all length scales. The surprisingly short dwell time observed in the crossover from ballistic to diffusive propagation is shown to be a consequence of the diffusive form of the energy density profile for the perfectly transmitting eigenchannel.

## Results

**Coherent vs. randomized waves in translucent and opaque samples**. The dominance of coherent or ballistic light in optically thin samples and of incoherent multiply-scattered light in opaque samples is illustrated in the recursive Green's function simulations[34], shown in Fig. 1. Simulations are carried out for a scalar wave of wavelength $\lambda_0 = 650$ nm propagating through a two-dimensional strip with reflecting sides along its length. A

random segment of length $L$ is sandwiched between regions of dielectric constant unity. The disordered region is divided into square elements with sides of length $\lambda_0/2\pi = 103.5$ nm and dielectric function $\varepsilon(x, y) = 1 + \delta\varepsilon(x, y)$ with $\delta\varepsilon(x, y)$ selected randomly from a uniform distribution in the range $-0.2$ and 0.2. The strip of width $W = 5.2$ μm supports $N = 16$ propagating waveguide modes. The $n = 1\ldots16$ waveguide modes have transverse profiles $\phi_n(y) \sim \sin(k^n_y y)$ with transverse components of the $k$-vectors $k^n_y = n\pi/W$ and longitudinal speed $v_n = ck^n_x/k$, where $c$ is the speed of light (details of the simulations are given in the Methods section).

In translucent samples, the transmission coefficient of coherent flux is of order unity, $|\langle t_{nn}\rangle|^2 \sim 1$, as seen in Fig. 1a. In contrast, the coherent flux in diffusive media is exponentially small, as seen in Fig. 1b. The coherent flux, $\langle t_{nn}(L)\rangle^2$, falls exponentially with sample length $L$ at different rates for each of the $N$ waveguide modes (Fig. 1c). However, the coherent flux falls at a single rate in the time domain, $1/\tau_s$, as seen in the inset of Fig. 1c. This yields the scattering mean free time and so the scattering mean free path, $\ell_s = c\tau_s = 27.2$ μm. Since the scale of the scattering element is much smaller than the wavelength, and fluctuations in $\varepsilon$ are small, $\ell_s$ is expected to be nearly equal to $\ell$[5].

The average delay time in transmission, $t_D$, which equals the average of the delay of the transmission channels weighted by the corresponding transmission eigenvalue, is shown in Fig. 1d (Supplementary Note 4). $t_D$ scales linearly for translucent samples and, in the thinnest samples, is equal to the average delay over all waveguide modes for a sample without disorder, $t_B = \langle L/v_n\rangle \equiv L/v_+$. Thus $v_+$ represents the average longitudinal component of velocity of a random incident wave. For the samples studied in simulations, $v_+ = 0.70c$. $t_D$ approach quadratic scaling for $L \gg \ell$.

**Scaling of optical transmission**. Since waves are largely coherent in translucent samples and randomized in diffusive media, one might expect the total transmission to scale differently in these regimes. Surprisingly, however, measurements of total optical transmission, which includes both the scattered and unscattered waves, were found to be in accord with diffusion theory down to sample lengths of $L \sim 2\ell$[8,13,15]. We explore wave propagation on still shorter length scales with $L \ll \ell$ to discover whether there is a lower limit in thickness below which the diffusion model fails. We note that computer simulations of the scaling of transmission of the portion of light that has been scattered at least once can be described by diffusion theory, even for $L \ll \ell$[14]. Here, however, we consider the full transmitted flux including light that has not been scattered, as is ordinarily the case in measurements of transmission.

For $L \gg \ell$, the scaling of average transmission of an incident beam is found by solving the diffusion equation with the impact of the boundary incorporated phenomenologically[13]. For a single incident channel $a$, the ensemble average of total transmission is $\langle T_a\rangle = (z_{p,a} + z_b)/(L + 2z_b)$[13] (Supplementary Eq. 7), where $z_{p,a}$ is the effective penetration depth of radiation in channel $a$ at which radiation is randomized and $z_b$ is the distance beyond the sample boundary in which the intensity within the sample extrapolates to zero. The model is solved for a randomized source at a depth $z_{p,a}$ with strength equal to the intensity that enters the sample. Surprisingly, the above expression is in excellent agreement with measurements down to $L = 2\ell$[13]. But one might not expect this model to apply to samples thinner than the penetration depth, since the effective source would then fall beyond the output boundary of the sample.

To explore transport in the crossover from ballistic to diffusive propagation, we measure the scaling of optical transmission through a dilute latex colloid contained in two wedge-shaped

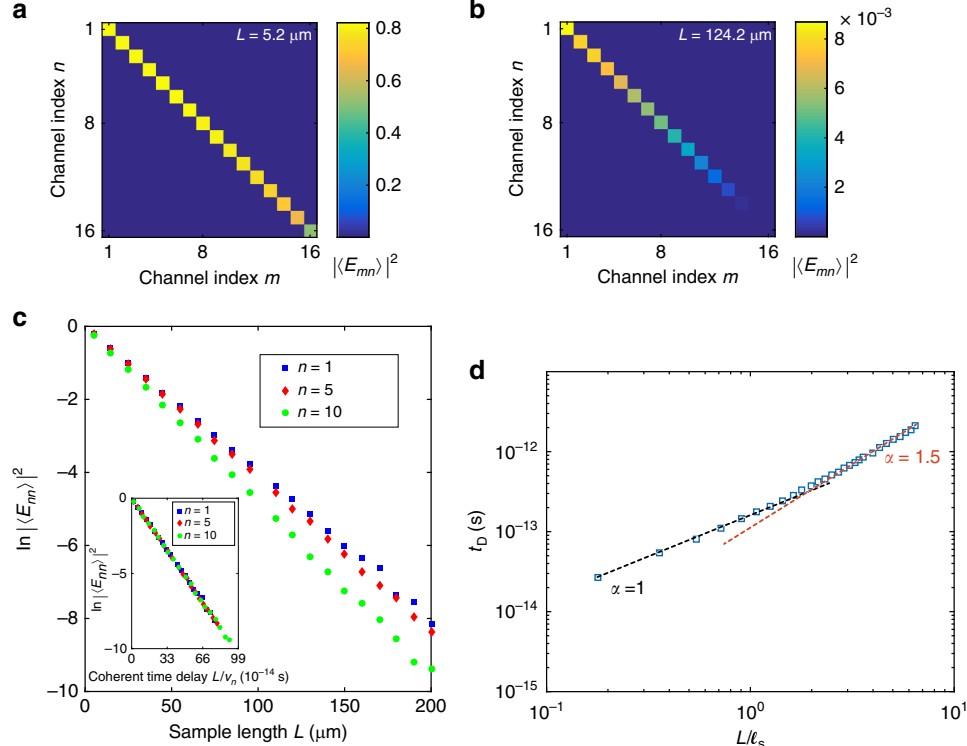

**Fig. 1** Simulations of wave transmission in opaque and translucent samples. **a**, **b** Average transmitted intensity $|\langle E_{mn}\rangle|^2$ for different incident and output waveguide modes. The coherent intensity, for $m = n$, is substantial for $L = 5.2\,\mu m$ and negligible for $L = 124.2\,\mu m$. **c** Scaling of the coherent intensity. The inset shows the scaling for three incident waveguide modes with $n = 1, 5, 10$. The variation of $|\langle E_{nn}\rangle|^2$ with coherent time delay $L/v_n$ for these modes with longitudinal velocities $v_n$ collapses to a single curve and falls exponentially to give a scattering length of $\ell_s = 27.2 \pm 0.2\,\mu m$. The scattering length is given by, $\ell_s = c\tau_s$, where $\tau_s$ is the mean free time obtained from the decay rate in the insert. **d** Log-log plot of the transmission delay time $t_D$ with sample length $L$. The dashed lines indicate different exponents $\alpha$ of the power law scaling. The transition from linear scaling occurs at $L \sim \ell$. As $L$ increases the value of $\alpha$ approaches 2

sample holders with different wedge angles. A normally incident laser beam is softly focused on the front of the sample while the transmitted light is collected in an integrating sphere (details of the optical measurements are given in Methods section). The thickness of the sample through which light passes is varied by translating the sample vertically perpendicular to the vertex of the wedge. The inverse of total transmission for the channel $a$ corresponding to the normally incident beam, $1/\langle T_a\rangle$, is seen in Fig. 2a to increase linearly with $L$ over the combined range of thicknesses in the two wedged samples of from $L = 20\,\mu m$ to 2.5 mm. From the distance beyond the sample of $2z_b$ at which $1/\langle T_a\rangle$ extrapolates to zero and the value of $2z_b/(z_{p,a} + z_b)$ to which $1/\langle T_a(L)\rangle$ extrapolates at $L = 0$, we obtain $z_b = 0.93\,mm$ and $z_{p,a} = 0.76\,mm$. This gives $\ell \sim 0.94\,mm$[13]. The linearity of measurements of $1/\langle T_a(L)\rangle$ from $0.05\ell$ to $2.7\ell$ shows that transmission follows the diffusion model even for $L \ll \ell$. Agreement of the scaling of transmission in the translucent regime with diffusion theory is also found in simulations in random 2D waveguides of the inverse of the total transmission averaged over all incident channels, $1/\langle T_a\rangle_a$, shown in Fig. 2b. Thus, despite the differences in propagation between translucent and opaque samples shown in Fig. 1, the expressions for the scaling of total transmission for a single incident channel (Fig. 2a) and for the average over all incident channels (Fig. 2b) apply equally in translucent and opaque media.

**Energy density distribution inside opaque and translucent media.** For diffusive waves, the flux though the sample is proportional to the spatial derivative of the energy density within the sample. It is of interest therefore to compare energy density profiles in samples thinner and thicker than $\ell$. Diffusion theory predicts a linear falloff of the average energy density with depth into a sample illuminated with a mixture of all incident waveguide modes. This is precisely what is found in the simulations shown in Fig. 2c for translucent as well as diffusive samples. Moreover, we find that the energy density extrapolates to zero at the same distance, $z_b = 19.2 \pm 0.2\,\mu m$ from the output surface for both opaque and translucent samples. This value of $z_b$ is in accord with the value found in simulations of the scaling of transmission shown in Fig. 2b of $z_b = 19.1 \pm 0.1\,\mu m$.

In Fig. 2c, we plot $W(x)$, the energy density integrated over the transverse direction at a depth $x$ averaged over random configurations and incident waveguide modes. $W(x)$ is normalized so that at it is equal to the average transmission coefficient through the sample at $x = L$, $W(L) = \langle T/N\rangle = u(L)v_+$. The transmittance $T$ is the sum over all channel-to-channel flux transmission coefficients, $T = \sum_{a,b=1}^{N}|t_{ba}|^2$, while $u(x)$ is the average energy density of a wave for unit incident flux.

The flux through a sample is given by Fick's first law of diffusion, $\langle T_a\rangle_a = -D\frac{du(x)}{dx}$, where $D$ is the diffusion coefficient. In two dimensional samples, $D = v\ell/2$, where $v$ is the speed of the wave. Since $W(x)$ extrapolates to zero at a distance $z_b$ beyond the output surface of the sample, we can show that $\ell = 2z_b v_+/v$ (Supplementary Eq. 4). This relation gives $\ell = 26.9 \pm 0.3\,\mu m$ which is close to the value of $\ell_s = 27.2 \pm 0.2$ found from Fig. 1c. Thus both transmission and the energy

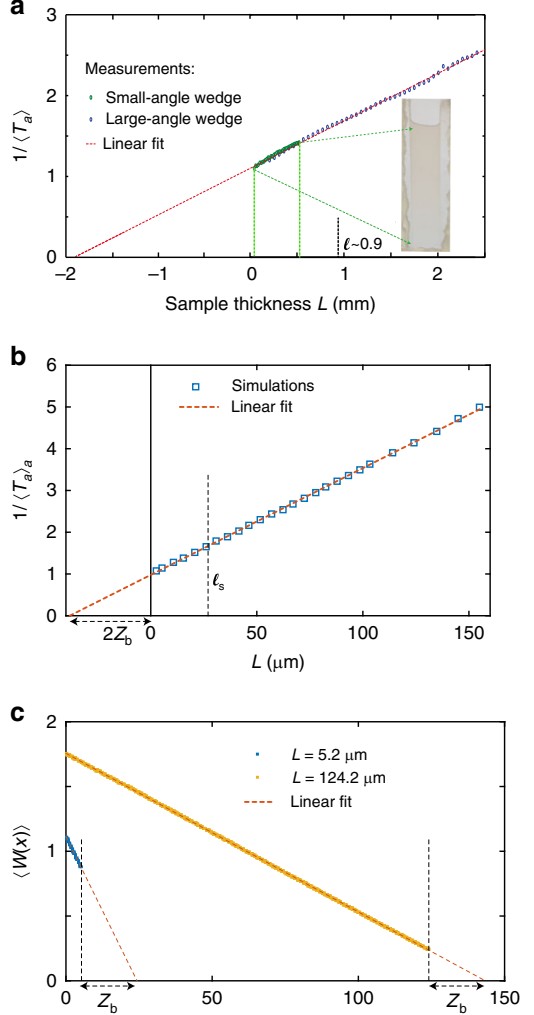

**Fig. 2** Scaling of total transmission. **a** Measurements of the scaling of the inverse of the optical transmission through a dilute suspension of 0.17-μm-diameter latex spheres in water. A photograph of the face of the translucent sample with a wedge angle of 0.86° is shown. The thicknesses at the beginning and end of the scan are indicated by the dashed green lines. The sides of the wedge are not shown because the microscope slides forming the faces of the sample are attached at their sides to a glass wedge with wax. The determination of $\ell \sim 0.9$ mm is discussed in the text. The value of $z_b$ is increased due to surface reflection at the air-glass interfaces. **b** Simulations of the scaling of the inverse of the total transmission averaged over all incident channels extrapolates to zero at $2z_b$, giving $z_b = 19.1 \pm 0.1$ μm. The vertical solid line indicates $L = 0$ μm and the vertical dashed line gives the value of $\ell_s$. **c** Results of simulations show a linear falloff of average energy density inside both translucent and multiple-scattering samples. The energy density extrapolates to zero beyond the sample boundary at the same distance $z_b = 19.2 \pm 0.2$ μm in both samples. The output boundaries of the two samples are indicated by the dashed vertical lines

density within the sample are well described by diffusion theory even in translucent samples.

**Transmission eigenvalues.** The scaling of conductance and transmission in multiply scattering media can be expressed in terms of the transmission eigenvalues, $\tau_n$. These are the ensemble averages of each of the $N$ eigenvalues of the $N \times N$ Hermitian matrix product $tt^\dagger$, where $t^\dagger$ is the Hermitian conjugate of the

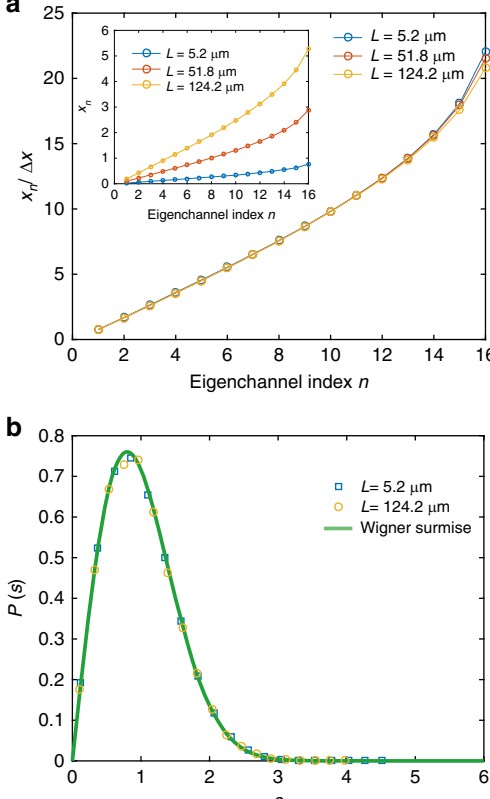

**Fig. 3** Structure of the transmission eigenvalues. The $x_n$ are determined by the transmission eigenvalues via the expression, $\tau_n = 1/\cosh^2 x_n$ with $x_n = L/\xi_n$. **a** The variation of the $x_n$ relative to the average spacing between them, $\Delta x$, are similar for different sample lengths, $L$. **b** The probability distribution of the spacing between the $x_n$ for $n < N/2$ in individual configurations is in accord with the Wigner surmise[27] for the eigenvalues of large random matrices for the Gaussian orthogonal ensemble, $P(s) = \frac{\pi}{2} e^{-\pi s^2/4}$ for both translucent and diffusive samples

transmission matrix $t$. The $\tau_n$ are indexed in order of decreasing transmission from $n = 1$ to $N$ and are proportional to the energy density on the output surface of the sample; their sum gives the average transmittance, $\langle T \rangle = \sum_1^N \tau_n$. The scaling of transmission eigenvalues, and, hence of the transmittance or conductance, was described by Dorokhov[22] in terms of a set of auxiliary localization lengths, $\xi_n$, where, $\tau_n = 1/\cosh^2 x_n$ with $x_n = L/\xi_n$. For $L \gg \ell$. The $x_n$ scale linearly for $n < N/2$ with spacing, $x_{n+1} - x_n \equiv \Delta x = L/\xi$, where $\xi = N\ell$ is the localization length. For $n > N/2$, the $x_n$ increase somewhat more rapidly[25,27].

Though waves in translucent samples are not randomized, the transmission matrix can still be defined and the scaling of the $x_n$ can be computed in simulations in the translucent as well as the diffusive regime. We find a common structure for the $x_n$ with the $x_n$ remaining equally spaced for $n < N/2$, as shown in Fig. 3a. The structure persists even in the thinnest samples for which the spacing is no longer proportional to $L/N\ell$ (Supplementary Fig. 3).

Another striking manifestation of universality is seen in the probability distributions of spacing between adjacent $x_n$ in different configurations normalized by the average spacing $= \Delta x$ for $n < N/2$. The distributions shown in Fig. 3b fall on a single curve corresponding to Wigner's surmise for the Gaussian orthogonal ensemble for eigenvalues of large random matrices[27]. This distribution, predicted for diffusive samples, is found to hold even for translucent samples. This reflects the universal repulsion between the $x_n$ seen in Fig. 3a and produces the same scaling law for transmission in translucent and diffusive samples.

**Transmission eigenchannels**. Since the similarity in the scaling of transmission in translucent and diffusive samples is related to the similarity in the statistics of the $x_n$, and so the $\tau_n$, it is interesting to explore whether there is a similarity in form between energy densities of the transmission eigenchannels in translucent and diffusive media. This will determine the energy density inside the sample, and ultimately the delay time in transmission[35–40] (Supplementary Eq. 10).

The transmission eigenchannels at the incident and output boundaries of the sample and the transmission eigenvalues are obtained from the singular value decomposition of the transmission matrix, $t$[27]. The field within the sample for the $n^{\text{th}}$ transmission eigenchannel cannot be obtained from $t$, but is just the field generated in the interior of the sample by the incident waveform for the transmission eigenchannels. We will consider $W_n(x)$ or $W_\tau(x)$, the contribution to $W(x)$ of the $n^{\text{th}}$ transmission eigenchannel or the eigenchannel with transmission $\tau$, which are normalized so that on the output surface, $W_n(L) = \tau_n$ or $W_\tau(L) = \tau$. The average profile of energy density throughout the sample excited by a mix of all incident channels is, $W(x) = \sum_1^N W_n(x)/N$, or equivalently an integral over the product of $W_\tau(x)$ and the probability density of $\tau$. To arrive at an expression for the functional form of the energy density profiles, it is useful to consider the scaling of the transmission eigenchannel profiles and to consider the profiles as functions of $x/L$, $W_\tau(x/L)$.

In diffusive samples, $W_\tau(x/L)$ can be written as the product of the profile of the completely transmitting eigenchannel with $\tau = 1$, $W_1(x/L)$, and a function $S_\tau(x/L)$, which is independent of $L/\ell$ and depends only on $\tau$, $W_\tau(x/L) = W_1(x/L)S_\tau(x/L)$[40]. $W_1(x/L)$ can be expressed as $1 + F_1(x/L)$, where $F_1(x/L) = A(L/\ell)[4(x/L)(1 - x/L)]$ is a solution of the diffusion equation with boundary conditions appropriate for perfect transmission[40]. $A(L/\ell)$ is the peak value of $F_1(x/L)$ at $x/L = 1/2$. We show in Fig. 4a and b that when $F_1(x/L)$ is normalized by its peak value, the curves for translucent and diffusive media collapse to the function $4(x/L)(1 - x/L)$. Thus, the spatial structure of the perfectly transmitting eigenchannel is the same in translucent and diffusive media.

We present results for $S_\tau(x/L)$ for $L/\ell = 0.18$ for three values of $\tau$ in Fig. 4c. We have not derived an expression for $S_\tau(x/L)$ for diffusing waves from first principles. However, the expression for transmission eigenvalues $\tau_n$ in terms of $x_n = L/\xi_n$ suggests a possible analytical expression for $S_\tau(x/L)$, which is in good agreement with the simulations in Fig. 4c. For a given value of $\tau$, the expression for $S_\tau(x/L)$ is an extension of Dorokhov's expression for $\tau_n$ on the surfaces of the sample into the interior[22]. The values of $S_\tau = W_\tau$ at $x = L$ and 0 of $\tau$ and $(2 - \tau)$, respectively, are consistent with the expression, $S_\tau(x/L) = 2\tau \cosh^2((1 - x/L)L/\xi') - \tau$, where $\tau$ is given by $1/\cosh^2(L/\xi')$. This expression matches the results of simulations in translucent samples for various values of $\tau$ shown in Fig. 4c. In diffusive samples, however, the expression above for $S_\tau(x)$ shows a systematic departure from simulations (Supplementary Fig. 5). Agreement with simulations in diffusive samples is only obtained once an empirical function is added in the argument of the hyperbolic cosine in the expression above for $S_\tau(x)$[40] (Supplementary Fig. 6).

A complete description of propagation in random media requires the scaling of the energy density profiles of transmission eigenchannels and so the scaling of $W_1(x/L)$. The form of the energy density for the completely transmitting eigenchannel, $W_1(x/L) = 1 + A(L/\ell)[4(x/L)(1 - x/L)]$ does not change throughout the translucent and diffusive regimes as seen in Fig. 4a and b. To find the scaling of $W_1(x/L)$, it remains to find the scaling of $A(L/\ell)$. The variation of the peak value of $W_1(x/L)$ with $L/\ell$ is

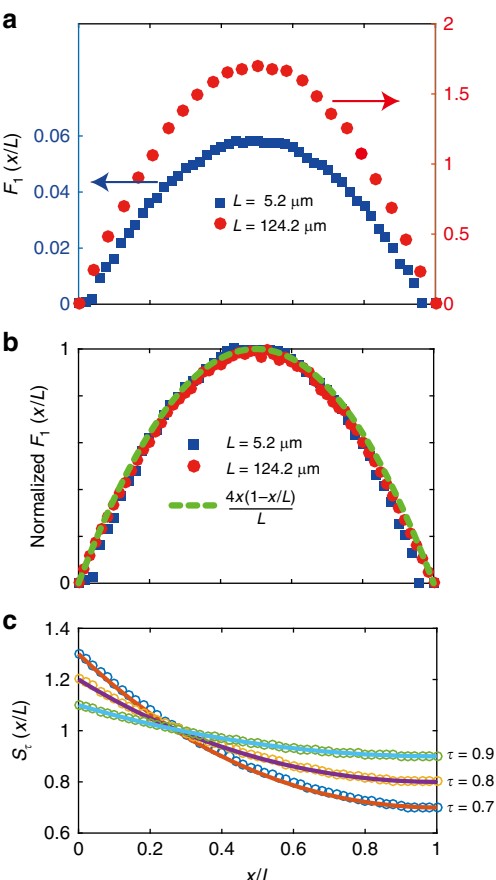

**Fig. 4** Profiles of energy density in transmission eigenchannels. **a** Profiles of completely transmitting eigenchannel in the translucent and diffusive regimes. **b** Profiles of $F_1(x/L) = W_1(x/L) - 1$ normalized by its peak value in the center of the sample for translucent and diffusive samples collapse to $4 \times (L - x)/L^2$. **c** Comparison of $S_\tau(x/L)$ found from simulations compared with the expression, $S_\tau(x/L) = 2\tau \cosh^2((1 - x/L)L/\xi') - \tau$, for values of $\tau = 0.7$, 0.8, and 0.9 in a sample with $L/\ell = 0.18$. $S_\tau(x/L)$ for small values of $\tau$ are not shown because they either do not occur, or occur too infrequently for good statistics to be collected for the energy density profiles

plotted in Fig. 5a and fit to the sum of a constant of unity and a linear term and a leading quadratic correction in $L/\ell$. The coefficient of the linear term is found to be 0.355.

Solving a generalized diffusion equation with flux at the output equal to the incident flux yields the peak value of $A(L/\ell) = v_+L/2v\ell$ (Supplementary Note 4). We have shown above that for our sample, the ratio of $v_+$ and $v$ is 0.7. This gives a linear contribution to $A(L/\ell)$ with coefficient 0.35, in agreement with the coefficient found in simulations. When $L$ approaches $\xi$, $A(L/\ell)$ is expected to increase more rapidly because coherent backscattering enhances the return of the wave to points in the medium[41]. Thus $W_1(x)$ is seen to be the sum of a constant "ballistic" term, a linear "diffusive" term, and "localization" correction that becomes important as $L$ approaches the localization length $N\ell$.

**Dwell times**. Measurements of optical[11,15,18,19,42] and ultrasound[16] pulsed transmission through random slabs show that on average photons arrive earlier than predicted by diffusion theory even in samples with $L > 5\ell$. The average delay time $t_D$ can also be determined from the transmission eigenvalues and energy density profiles of the transmission eigenchannels[37] (Supplementary Note 4).

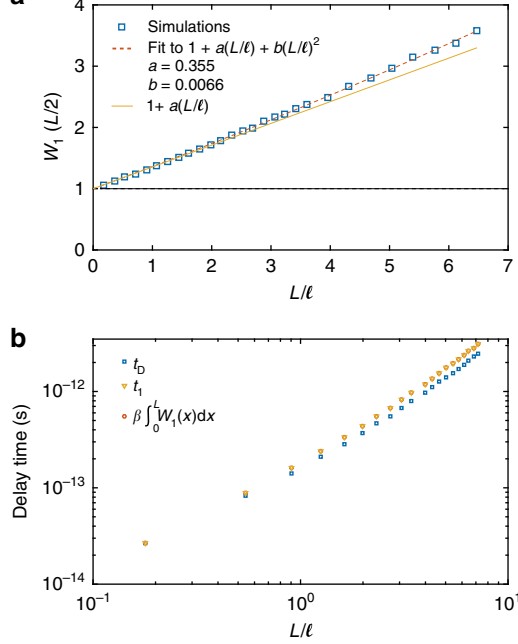

**Fig. 5** Scaling of $W_1(L/2)$ and the delay times. **a** The scaling of $W_1(L/2)$ (blue squares) is fit by a parabolic function $1 + a(L/\ell) + b(L/\ell)^2$. The fit gives $a = 0.355$ and $b = 0.0066$ (red dashed curve). The linear coefficient $a$ can be calculated using diffusion theory, while the quadratic term reflects enhanced delay due to incipient localization. The sum of the constant term of unity (black dashed horizontal line) and the linear term of $a(L/\ell)$ is shown as the yellow solid line. **b** The delay time of the fully transmitting eigenchannel obtained from the composite phase derivative of the eigenchannel with respect to the frequency shift[38] is shown as the triangles in Fig. 5b. The integral of $W_1(x)$ multiplied by the proportionality constant $\beta$ is shown as the red circles in the log-log plot of Fig. 5b. The overlap of the two plots shows that the integral $W_1(x)$ is proportional to the delay time of the fully transmitting eigenchannel (Supplementary Note 4). The scaling of $t_D$, shown as the blue squares, is similar to the scaling of $t_1$ for diffusive waves

It can be expressed as the average delay time of the transmission eigenchannels $t_n$ weighted by the corresponding transmission eigenvalues, $\tau_n$, $t_D = \sum_1^N \tau_n t_n / \sum_1^N \tau_n$. The eigenchannel delay time is proportional to the energy stored within the sample so that $t_n \sim \int_0^L W_n(x)\mathrm{d}x$[37] (Supplementary Note 4).

In Fig. 5b, we plot $t_D$ and the delay time of the fully transmitting eigenchannel, $t_1$. Since the form of $S_\tau(x)$ is independent of $L/\ell$ for diffusive waves, the scaling of $t_D$ for $N > L/\ell > 1$ largely depends upon the scaling of $t_1$, which is given by the integral of $W_1(x)$ over the sample length. Only for $L/\ell = 2.65$ is the amplitude of the "diffusive" component of $W_1(x/L)$, equal to the value of the "ballistic" component, while the value of the integral of the diffusive term over the sample length only reaches that for the ballistic term for $L/\ell = 3.82$. In addition to the small slope of $A(L/\ell)$ vs. $L/\ell$, the dwell time increases slowly in thin samples because the superlinear increases of the integral of $W_1(x)$ (Supplementary Eq. 10) is offset by the sublinear increases of the $t_n$ (Supplementary Fig. 7). In contrast, for thicker samples, $\tau_n$ is typically small for channels $n > g$ so that low transmission eigenchannels do not contribute appreciably to $t_D$ (Supplementary Fig. 7). For these reasons, the onset of diffusive scaling of the dwell time only begins when $L/\ell$ is substantially larger then unity. Thus, it is precisely the similarities in the functional form of characteristics of static transport between translucent and opaque

samples which lead to reduced delay times relative to predictions of the diffusion model.

The shorter delay time in transmission relative to diffusion theory[11] limits the time in which the wave can spread in the transverse direction and so results in a reduced width of the transverse profile of intensity on the output surface in thin samples[13] and early times[18] relative to diffusion theory. In thicker strong scattering samples, observations of a halt in the transverse spread of the intensity profile on the output surface indicate that the wave is localized[43]. Though the present study has focused on longitudinal propagation in translucent and diffusive quasi-one-dimensional samples, the evolution of the transverse intensity distribution with sample thickness in samples of any scattering strength can be studied in the slab geometry within the framework of transmission eigenchannels by decomposing a narrow incident beam into a sum of transmission eigenchannels.

## Discussion

A consistent picture of propagation in the crossover from ballistic to multiple scattering has long remained elusive. On the one hand, the scaling of transmission in samples hardly thicker than a mean free path still obeys diffusion theory, while on the other the dwell time in samples up to several times the mean free path scale only slightly faster than linearly, as would be expected for waves following nearly ballistic trajectories. This work shows that the questions raised are even more perplexing since measurements of optical transmission are found to scale diffusively down to one-fiftieth of the mean free path.

We show here that a description of the energy density and flow within random translucent and opaque systems emerges from the common statistics of the ratios of the sample length and eigenchannel localization lengths, $x_n = L/\xi_n$, together with the intensity profiles of the associated transmission eigenchannels. Transmission is determined by the sum over transmission eigenvalues, which reflects the mutual repulsion of the $x_n$, while the deviation of dwell time from diffusion theory is a consequence of the diffusive form of the energy density profiles of transmission eigenchannels even in translucent samples. The delay time for diffusive samples is largely determined by the profile of the fully transmitting transmission eigenchannel $W_1(x/L)$, which includes a factor which is the sum of a constant ballistic term, a diffusive term linear in $L/\ell$, and a leading-order localization correction which is quadratic in $L/\ell$. It is the small coefficient of the linear term relative to unity which is largely responsible for the slow approach to the quadratic scaling of $t_D$ associated with diffusion.

The delay time in reflection, which is of importance in optical or ultrasound diffuse tomography, can also be given in terms of the properties of transmission eigenchannels. Since the delay time of transmission eigenchannels is the same in reflection as in transmission[37] and the reflection coefficient in the $n^{\text{th}}$ transmission eigenchannel is $(1 - \tau_n)$, the average delay time in reflection is $t_D^{\text{reflection}} = \sum_1^N (1 - \tau_n) t_n / \sum_1^N \tau_n$[37].

The work in this paper opens the door for study of many open issues. Among these are a fuller expression for the localization contribution to $W_1(x/L)$, not only the coefficient of the normalized function $F_1(x/L)/F_1(1/2)$, but also the deviation of this function from the diffusive form. If propagation is primarily through single peaked localized states, one would expect that $F_1(x/L)/F_1(1/2)$ would narrow significantly since the intensity should be peaked within a localization length of the center of the sample for high maximal transmission[44]. But if the width of this function does not change appreciably, transport would then largely be through coupled localization centers, known as necklace states, in which the incident wave is coupled strongly through the sample[45]. Thus, the width of $F_1(x/$

$L)/F_1(1/2)$ would indicate the dominance of the transport through either isolated states or necklace states for localized waves. The existence of both single peaked localized states and multiply peaked necklace states has been observed in layered media[46], single mode waveguides[47], natural materials[48], and can be created in multimode optical fiber with mode coupling[49]. It is also of great interest to explore the disposition of energy within thin anisotropic scattering media, of importance in biomedical research[50].

Obtaining the mean free path over the full range of opacity is also of importance in monitoring colloidal, micellar, or metallic nanoparticle concentrations, sedimentation, atmospheric conditions, and medical diagnostics. Since the scaling of transmission and time delay depend on $\ell$ and $z_b$ in different ways, the results presented here suggest that it should be possible to determine the mean free path in samples over a broad range of $L/\ell$. In future work the relationship between $\ell$ and $z_b$ in the presence of internal reflection will be determined in the regime of the crossover from translucent to multiply scattering samples. These results would, for example, provide a path towards quantitative monitoring of particulate concentrations in liquids or gases in sample with thickness of the order of the mean free path. The transport mean free path can also be obtained from the spacing of the $x_n$ in translucent samples, in which the measurements of the TM can be more complete since the number of coherence areas is relatively small in translucent media[31].

Recent developments of techniques for measuring the transmission matrix for imaging applications are relevant to both thin and thick scattering samples. A clearer picture of the connection between energy density and time delay in scattering are of importance in many approaches to imaging. For example, in medical imaging, different regions of a sample are probed in diffusing temporal field correlation spectroscopy[51] as the distance between the probe and source are changed, while different dwell times within the medium may be probed even for fixed spacing by utilizing correlation spectroscopy in the time domain[52]. These techniques are important in non-invasively monitoring blood flow and managing the delivery of oxygen to the brain.

## Methods

**Numerical simulations of a scalar wave propagating**. The Green's function $G(\mathbf{r},\mathbf{r}')$ between arrays of points on the input surface $\mathbf{r} = (0, y)$ and at a depth $x$, $\mathbf{r}' = (x, y)$ can be obtained by solving the wave equation $\nabla^2 E(x,y) + k_0^2 \varepsilon(x,y) E(x,y) = 0$ on a square grid via the recursive Green's function method. To calculate the transmitted flux for various incident and output waveguide modes, the Green's function is expressed in terms of the basis of the waveguide modes, $t_{ba}(x) = \sqrt{\nu_b \nu_a} \int_0^W dy' \int_0^W dy \, \phi_b(y) \phi_a^*(y') G(\mathbf{r}, \mathbf{r}')$, in which $\nu_a$ is the group velocity of the waveguide mode $a$, and $W$ is the width of the waveguide.

The incident wavefront $\mathbf{v}_n$ and outgoing filed $\mathbf{u}_n$ associated with the $n^{th}$ eigenchannel can be found using the singular value decomposition of the transmission matrix, $t = U\Lambda V^+$, where $\mathbf{u}_n$ and $\mathbf{v}_n$ are columns of the unitary matrix $U$ and $V$, respectively. $\Lambda$ is a diagonal matrix with elements $\sqrt{\tau_n}$. The field at a depth $x$ for an incoming eigenchannel in momentum space is found by multiplying the transmission matrix $t_{ba}(x)$ by $\mathbf{v}_n$. Summing the square of the coefficients over the $N$ waveguide modes yields the density of the flux at $x$. At the output surface, $x = L$, this gives $\tau_n$. The energy density $W_n(x)$ can then be obtained by dividing the density of the flux by the average speed $\nu_+$ of the wave propagating through the waveguide. The scaling of the total transmission shown in Fig. 2b was obtained by averaging over 5000 sample configurations. $W_n(x)$ for $L = 5.2\,\mu m$ and $124.2\,\mu m$ was averaged over 200,000 and 10,000 samples, respectively, and the energy distributions for eigenchannels with a specific value of transmission $\tau$ are found by averaging the eigenchannel with transmission between $0.98\tau$ and $1.02\tau$. To find the scaling of the peak value of the $F_1(x)$, 500 sample realizations were averaged for each of the lengths of samples to ranging from 5.2 to 154.5 $\mu m$ to yield the $\langle W_1(x) \rangle$. The profile of the fully transmitting eigenchannels for $\tau > 0.98$ was subsequently fitted with a parabolic function to give the peak value.

**Optical measurements of light propagation through a wedged random medium**. The scaling of total transmission is measured for a colloid of 0.17-μm-diameter polystyrene spheres in water at a volume fraction of ~0.003. An anionic surfactant was added to the colloidal suspension to prevent particle aggregation. The latex spheres and surfactant were obtained from Polysciences. The colloid is placed in two wedged sample holders made from microscope slides meeting at vertex angles of $\theta_{wedge} = 0.86°$ and $5.88°$. Polished glass and aluminum wedges were used as spacers between the slides. The sides of the assembly were sealed with wax. The normally incident beam of light at 532 nm is weakly focused on the incident face of the sample. The sample is translated perpendicular to the vertex line in steps of 1 mm after each measurement of transmission. The light spreads to a spot on the output plane with diameter of order of $L$. Because the wedge angles are small, the variation in thickness $L$ of the colloid across the illuminated region of the sample is much smaller than the sample thickness $L$. The transmitted light is collected in a Labsphere integrating sphere.

**Data availability**. The authors declare that all data that support the findings of this study are available from Zhou Shi at zhoushi.qc@gmail.com upon reasonable request.

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

## Acknowledgements

We thank X. Cheng for help with the optical measurements and acknowledge useful discussions with V. A. Gopar, X. Cheng, M. Davy, and C. Tian. This research was supported by the National Science Foundation (DMR-BSF: 1609218).

## Author contributions

Z.S. carried out the numerical simulations. A.Z.G. performed the optical measurements and the calculations. A.Z.G. and Z.S. wrote the manuscript.

## Additional information

**Competing interests:** The authors declare no competing interests.

