## [Peer Review File · Nature Communications]

Reviewers' Comments:

Reviewer #1:

Remarks to the Author:

I believe that also in this new version the paper delves into an interesting topic and presents some surprising results about light diffusion in semi transparent media.

As before, I believe the paper is suitable for publication in one of the Nature Journals.

Reviewer #2:

Remarks to the Author:

Although I find this work a bit technical for Nature Communications and probably difficult to follow for non-specialists, the authors nearly convince me of its potential impact in wave physics. The fact that some diffusive result holds for very thin samples can be of interest in optical or ultrasound diffuse tomography.

I have two remarks:

1/ On sentence puzzles me in the rebuttal letter: "The paper also shows that the cause of the reduced transverse spread of incident flux compared to assumed diffusive results is a consequence of the shorter delay time in transmission than had been previously expected".

I don't see in the paper any discussion about the transverse spread of the incident flux, whereas it can be an important result since a lot of experiments now investigate the spatial growth of the diffusive halo:

- Pattelli, L., Mazzamuto, G., Wiersma, D. S., & Toninelli, C. Phys. Rev. A 94, 043846 (2016).

- L. Pattelli, R. Savo, M. Burrelli, and D. S. Wiersma, Light Sci. Appl. 5, e16090 (2016).

- A. Badon, D. Li, G. Lerosey, A. C. Boccara, M. Fink, A. Aubry, Optica 3, 1160-1166 (2016)

- T. Sperling, W. Bührer, C. M. Aegerter, and G. Maret, Nat. Photonics 7, 48–52 (2012)

Among these works, the first one investigates the same problem as in the current paper. I think that would be great if the authors could discuss about that paper and about the transverse growth of the diffusive halo and how it connects to the delay time in transmission.

2/ I think the authors should also discuss about reflection measurements in the paper. In a lot of in-vivo experiments, we only have access to reflection measurements. What is the scaling of the static/dynamic reflected intensity for small optical thickness?

2/ The authors are talking about a leading order localization correction which is quadratic for $W_1(L/2)$? How do they know it is quadratic? From numerical simulations (Fig.5a)? Is there any proof of that?

If the authors satisfactorily take into account my remarks (mainly 1 and 2), I would favor publication in Nature Communications.

Reviewer #3:

Remarks to the Author:

This paper has gone through several rounds of revisions, and I think it is suitable for publication in Nature Communications in its present form.

Response to the Reviewers

We thank the reviewers for their continuing engagement and patience with our manuscripts. We are gratified that they have seen merit in the demonstration of diffusive characteristics of propagation in translucent samples and the unifying framework of transmission eigenchannels.

We appreciate the judgement of Reviewers #1 and #3 that the work can be published as it is.

Referee #2 notes that the work is a bit technical. We believe that, in time, that as the concept of transmission eigenchannels becomes more familiar, the sequence of results and logic of the paper will be more accessible to non-specialists. We thank the reviewer for noting the relevance of the results to optical and ultrasound tomography. This is now mentioned in the Conclusion section.

I/ One sentence puzzles me in the rebuttal letter: "The paper also shows that the cause of the reduced transverse spread of incident flux compared to assumed diffusive results is a consequence of the shorter delay time in transmission than had been previously expected". I don't see in the paper any discussion about the transverse spread of the incident flux, whereas it can be an important result since a lot of experiments now investigate the spatial growth of the diffusive halo:

- Pattelli, L., Mazzamuto, G., Wiersma, D. S., & Toninelli, C. Phys. Rev. A 94, 043846 (2016).

- L. Pattelli, R. Savo, M. Burrese, and D. S. Wiersma, Light Sci. Appl. 5, e16090 (2016).

- A. Badon, D. Li, G. Lerosey, A. C. Boccarda, M. Fink, A. Aubry, Optica 3, 1160-1166 (2016)

- T. Sperling, W. Bührer, C. M. Aegerter, and G. Maret, Nat. Photonics 7, 48–52 (2012)

Among these works, the first one investigates the same problem as in the current paper. I think that would be great if the authors could discuss about that paper and about the transverse growth of the diffusive halo and how it connects to the delay time in transmission.

The idea quoted by the reviewer from our rebuttal letter regarding reduced transverse spread of incident flux compared to assumed diffusive results is now discussed in the paper in the last paragraph of the Discussion section:

“The shorter delay time in transmission relative to diffusion theory ~~is consistent with the~~ results in a reduced width of the transverse profile of intensity on the output surface in thin samples¹³ and early times¹⁸ relative to diffusion theory. In thicker strong scattering samples, observations of a halt in the transverse spread of the intensity profile on the output surface indicate that the wave is localized⁴³. Though the present study has focused on longitudinal propagation in translucent and diffusive samples, the transverse evolution of the intensity distribution with sample thickness in samples of any scattering strength can be studied within the framework of transmission eigenchannels by considering the intensity profiles of transmission

eigenchannels in a slab illuminated by a narrow beam. The transverse spread would be different for each transmission eigenchannel, and would be expected to increase with t_n . The average transverse intensity profile would be given by the sum over all transmission eigenchannels.”

We believe that a fuller discussion of this point would add complexity and go beyond the scope of the present paper, which treats steady-state propagation. Even the dwell time relates to average delay at a single frequency. But describing an approach based on transmission eigenchannels that could be taken to find the profile of transmitted intensity is in line with the thrust of the paper. Obtaining the full time dependence would be more complex since this would need to treat the spectrum of transmission eigenchannels.

We have included the references noted by the reviewer that were not previously referenced, as these were relevant to the issue raised of the transverse intensity profile.

2/ I think the authors should also discuss about reflection measurements in the paper. In a lot of in-vivo experiments, we only have access to reflection measurements. What is the scaling of the static/dynamic reflected intensity for small optical thickness?

We now give in the Conclusion the expression for the scaling of the delay time in reflectance in terms of the transmission eigenchannel, as suggested by the referee.

“The delay time in reflection, which is of importance optical or ultrasound diffuse tomography, can also be given in terms of the properties of transmission eigenchannels. Since the delay time of transmission eigenchannels is the same in reflection as in transmission³⁷ and the reflection coefficient in the n^{th} transmission eigenchannel is $(1-\tau_n)$, the average delay time in reflection is $t_D^{\text{reflection}} = \sum_1^N (1-\tau_n)t_n / \sum_1^N \tau_n$. ”

2/ The authors are talking about a leading order localization correction which is quadratic for $W1(L/2)$? How do they know it is quadratic? From numerical simulations (Fig.5a)? Is there any proof of that?

2/ The authors are talking about a leading order localization correction which is quadratic for $W1(L/2)$? How do they know it is quadratic? From numerical simulations (Fig.5a)? Is there any proof of that?

In Fig. 5a, we fit $W1(L/2)$ to second order in the ratio of the sample length and mean free path, L/ℓ . The constant and the first order term in L/ℓ are found to correspond precisely to the prediction of diffusion theory. We believe therefore that the quadratic additional term must be

due to effects associated with the breakdown of diffusion and the onset of localization; there does not seem to be anything else it could be. Calculating the scaling of $W_1(L/2)$ and the shape of $W_1(x)$ for localized waves are important challenges that we hope to pursue in future work. However, they fall outside the scope of this work which explores the nature of transport of ballistic and diffusive waves. We believe treating additional subjects would make the paper more complete, but would overload the paper and make it difficult to read.

We hope that with the changes made the paper can now be judged to be suitable for publication.

Sincerely,
Azriel Genack and Zhou Shi

Reviewers' Comments:

Reviewer #2:

Remarks to the Author:

The authors have satisfactorily addressed my remarks. I suggest publication of this paper in Nature Communications.